# An Empirical Analysis of Static Analysis Methods for Detection and Mitigation of Code Library Hallucinations

## Abstract

Despite extensive research, Large Language Models continue to hallucinate when generating code, particularly when using libraries. On NL-to-code benchmarks that require library use, we find that LLMs generate code that uses non-existent library features in 8.1-40% of responses. One intuitive approach for detection and mitigation of hallucinations is static analysis. In this paper, we analyse the potential of static analysis tools, both in terms of what they can solve and what they cannot. We find that static analysis tools can detect 16-70% of all errors, and 14-85% of library hallucinations, with performance varying by LLM and dataset. Through manual analysis, we identify cases a static method could not plausibly catch, which gives an upper bound on their potential from 48.5% to 77%. Overall, we show that static analysis methods are cheap method for addressing some forms of hallucination, and we quantify how far short of solving the problem they will always be.

## 1 Introduction

LLMs can hallucinate arguments to functions and even entire functions. Spracklen et al. (2025) found that GPT-4 turbo hallucinated packages 4% of the time, while CodeLlama 7B hallucinated them 26% of the time. Tian et al. (2025) found that GPT-4 mapped data types and structures incorrectly 10% of the time, and its output did not match external knowledge sources, such as modules' imports, in 0.6% of cases. Identifying and fixing these errors creates additional work for programmers (Tanzil et al., 2024). If the errors are missed it can create a security risk, with an attacker who notices a common hallucination creating a malicious package that matches it (Spracklen et al., 2025; Krishna et al., 2025). Prior work has applied static analysis tools to detect syntax errors and logical errors (Ding et al., 2023; Ugare et al., 2024; Poesia et al., 2022), but not hallucinations.

We analyse the potential for static analysis to detect and mitigate hallucinations in coding output that involves library usage. We consider both errors, i.e., any bug that leads to incorrect code behaviour, and hallucinations, i.e., code that would work if imagined functions or arguments existed.

We consider three static analysis methods, applied to the output of six LLMs on three benchmarks. Two methods are off-the-shelf static analysis tools: Mypy and Pyright. One is a grammar that we automatically constructed from docstrings of libraries. All three can be applied after generation, to detect hallucination. The grammar can also be used during generation to constrain decoding. We also compare to prompting o3-mini as a baseline. We evaluate both detection and mitigation using three NL-to-code benchmarks that involve library use: DS-1000, Odex, and BigCodeBench.

While static analysis tools provide more coverage of bugs (up to 70%), grammars successfully identify some imaginary features (up to 15%). We conduct manual analysis to establish an upper bound on the performance of static analysis methods in a type-inferred language (Python) and identify which error categories cannot be solved by these approaches. Our results also indicate the importance of code benchmarks that require library use and have clear NL requests, as otherwise, this form of hallucination will be missed.

The contributions of this paper are:
(1) A comprehensive analysis of various static tools for detecting and mitigating library hallucinations,

(2) Manual annotations from code completions with labels on hallucinations of three NL-to-code benchmarks on one open-source and three closed models, and

(3) A framework for inspecting docstrings and transforming those into grammar form for constrained decoding on open-domain code.

## 2 Related work

**Hallucinations in Code**  Tian et al. (2025) define code hallucination as code generated by LLMs that might be syntactically and semantically plausible but cannot execute or meet requirements. They consider code errors as a type of code hallucination. Our definition, described in the next section, is slightly different. We do not consider all errors to be hallucinations. Existing work in AI on addressing code hallucinations uses approaches such as retrieval-augmented generation (RAG), iterative grounding, few shot prompting, and fine-tuning (Eghbali & Pradel, 2024; Liu et al., 2024a; Agarwal et al., 2024; Mok et al., 2024; Li et al., 2023). Although these approaches sample better quality tokens, they differ from our work by not addressing detection and mitigation simultaneously (Tanzil et al., 2024), and cannot guarantee that hallucinations have been resolved. There has been related work in Software Engineering on Automatic Program Repair (APR), but this is a more general challenge (Xia et al., 2023; Koutcheme et al., 2023; Prenner et al., 2022; Liu et al., 2024b; Zhang et al., 2023; Wuisang et al., 2023; Ni et al., 2024). Our research aims to mitigate hallucinations by detecting and constraining them before the code is executed and has an error.

**Hallucinations and External Knowledge Sources in Code**  LLMs pre-trained on specific tools produce inconsistencies and hallucinations in APIs (Roy et al., 2024), and LLMs optimised for code produced higher rates of package hallucinations (Krishna et al., 2025). Ayala & Bechard (2024) explored finetuning on JSON-structured workflows with annotated suggestions. When they compared a model with and without augmentation, they observed a loss in diversity and an increase in hallucinations. Eghbali & Pradel (2024) proposed a RAG approach for factual open-domain code, which iteratively queries an LLM with API references as context. RAG reduced hallucinations by 3%, and the iterative approach led to a 15% improvement in matching the exact imports, though they did not confirm that the code was executable.

**Static Analysis in Code Generation**  Ding et al. (2023) suggest that static program analysis from generated code can reduce errors and resources when evaluating code generation with execution-based benchmarks. Jaoua et al. (2025) showed that RAG with knowledge bases and static code analysis can be a cost-efficient method for code reviews. However, in the context of bug detection, Chen et al. (2025a) compared static analysis tools with finetuned and pre-trained bug detection models. They found that when the tool and the model were used together, recall increased, and precision also increased on annotated programs. This shows the importance of well-annotated packages for accurate static analysis. We relied on static code analysis to prevent faulty code generation, while instructing the LLM to repair bugs detected by these tools.

**Inference Based Solutions**  In package hallucination, Spracklen et al. (2025) demonstrated that hallucinations are not due to sampling, as they still occur with greedy inference. Fu et al. (2024) used constrained decoding to produce more secure code, which is different from our goal, but supports the feasibility of our approach. Roy et al. (2024) perform constrained decoding to check a conversation intent and factuality in self-created APIs, but not in real-world use. Chen et al. (2025b) tested API use, but did not evaluate execution, nor requests from natural language. While they use library aliases as prefixes, when compared to our approach, using a grammar can provide additional validation on syntax and semantics.

**Grammar-Constrained Decoding**  Focusing on errors, Olausson et al. (2023) concluded that the effective rate of code repair directly relates to the quality and size of the LLM. In contrast, Geng et al. (2023) found that grammar-constrained decoding, without finetuning and with scarce data, boosts any size of LLM on structured tasks defined through formal grammar. Ugare et al. (2024) used context-free grammars to address syntax errors, reducing them by 96% on Python and Go, which clearly demonstrates the potential of this approach to resolve issues in output. Similar to our work, SYNCHROMESH (Poesia et al., 2022) explored grammar constraints on SQL and JSON output, derived from samples. While flexible, that has the disadvantage that it is not possible to guarantee consistency with an external library.

**Benchmarks** Benchmarks on code hallucinations (Hallucode; Liu et al. 2024a; CodeMirage; Agarwal et al. 2024) treat this problem as a classification task, where the language model needs to detect the type of hallucination from a given snippet of code. They tend to focus on logical errors or inconsistency with the request, whereas we are interested in library use, leading us to evaluate with open-domain code execution benchmarks.

CodeRag-Bench (Wang et al., 2025) evaluated the use of RAG to improve accuracy in several NL-to-code benchmarks. CodeRag-Bench focused on Python benchmarks, as it is the most widely used language for code generation. This popularity contributes to a higher number of benchmarks compared to other programming languages, which limits the analysis outside of Python.

## 3 Defining Code Hallucination

LLMs can make a variety of errors in code generation. We aim to be consistent with the definition of a hallucination from the 'Speech and Language Processing' textbook by Dan Jurafsky and James H. Martin: "A hallucination is a response that is not faithful to the facts of the world. That is, when asked questions, large language models sometimes make up answers that sound reasonable" (Jurafsky & Martin, 2024).

We apply this definition to code by treating the 'facts of the world' as existing libraries, APIs, or user-provided context. Therefore, library hallucinations include: non-existent libraries (e.g. Module Not Found Errors), generating a function name that does not exist in a library (e.g. Attribute Errors or Import Errors), nonexistent parameters in functions, or mapping incorrect data types and misrepresenting object interfaces explicitly defined in the library (e.g. Type Errors) as they violate external factual knowledge. Note, however, that not all errors are hallucinations. For example, a syntax error in code is comparable to a grammatical error in language: a violation of internal language constraints.

## 4 Detection Methods Evaluated

To detect hallucinations, we consider three types of approaches: (1) using a grammar that checks library calls, (2) off-the-shelf static code analysis tools, and (3) a simple LLM-as-a-judge baseline. To mitigate the errors these tools identify, we instruct an LLM to repair the code using the tool's analysis output as guidance. For the grammar based method, we also consider constrained decoding as a mitigation strategy.

### 4.1 Automatically Extracted Grammars

In this approach, a GBNF[1] grammar defines whether code is consistent with library definitions. This can be used either with grammar-constrained decoding, or with post-generation analysis. The grammar contains the core language definition and additional symbols to cover the contents of libraries:

**Except-library**: accepts any token aside from the library's alias and name, and the keywords "import" and "from".

**Library**: uses the library name or alias as a prefix for built-ins, initialisers, and constants.

**Import**: uses the keywords "import " and "from " as a prefix to validate packages in the environment of the benchmark and Python built-in modules.

These allow us to handle any output not related to libraries, including code and even natural language. This is useful as it allows us to focus on identifying issues with library use. Also, it allows for the style of output where there is a brief explanation, followed by a code snippet, and then further explanation in NL.

We construct the rest of our grammar in two steps. First, we use the *inspect* module to retrieve the target library's documentation and represent the key information in JSON. Second, we combine information from the JSON data with Python's core language specification to create a single GBNF grammar that covers

---

[1]This is a variant of Extended Backus-Naur Form (EBNF) developed for llama.cpp that adds some features from regular expressions and can be directly used for constraint decoding.

library use in the context of other code. Each rule accounts for one of the library functions, with variations to handle optional and repeated arguments.

To create grammars for additional libraries, all that is required are the library's docstrings and a list of common aliases (e.g., `np` for numpy). We constructed our list of common aliases automatically based on how they are used in the benchmarks we consider. In this work, we focus on Python due to its widespread use and the prevalence of Python-based benchmarks.

To extend to other languages would require two practical steps. First, the target language's formal grammar would need to be manually converted into GBNF format, a process that took approximately 4 hours for Python. Second, the Schema-to-Grammar conversion script would need to be adapted for the target language, though these changes are minimal as the JSON Schema representation of library grammars is language-agnostic, meaning libraries already captured in JSON Schema do not need to be re-inspected. Only language-specific libraries that have not yet been retrieved would require an equivalent inspection package. For additional information and examples about the construction of the grammar, please see Appendix B.

**Grammar-constrained decoding** We use the grammar to constrain the output of the LLM to be consistent with what the grammar permits. During inference, we constrain the probability distribution of the next token to only have positive values for tokens that are valid according to our grammar. Which tokens can be positive is determined by a parser that matches partial sequences with the grammar we provide.

We use the parser and decoding integration provided by *llama.cpp*. This has the advantage that our approach is compatible with any of the open-source LLMs hosted in Ollama. To be efficient, *llama.cpp* actually samples without filtering the space of output tokens, and if the sample is accepted by the grammar it continues without needing to compute the mask over the output token options. To further improve speed, we implemented caching the states from the non-deterministic pushdown automata used by *llama.cpp*.

## 4.2 Off-the-Shelf Analysers

Our second approach is to use off-the-shelf Static Code Analysis tools. Specifically, we consider *mypy* (an explicit type annotation tool) and *pyright* (a type inference tool). These were developed to detect errors in human-written Python code. They are more general than our grammar approach, but are also more language-specific. We configured both tools with automatically generated type stubs to provide information about the functions in the standard library and third-party libraries. Providing these annotations is critical for identifying the library features for our use case.

## 4.3 LLM-as-a-judge

As a baseline alternative, we ask o3-mini to judge whether code will be executable. To do so, we provided the generated code in conjunction with a closed-answer question to the model. We present an example of this prompt and LLM's response in the Appendix D.2. This approach could be considered a static analysis method since the LLM does not execute the code. However, it does not have the low overhead or behaviour guarantees of traditional static analysis methods.

## 5 Experiments

**Benchmarks** Recent benchmarks developed to target hallucinations in code (Collu-Bench; Jiang et al. 2024; CodeHalu; Tian et al. 2025, APIHulBench; Chen et al. 2025b) either lack NL descriptions or evaluate with an exact match, rather than using test cases. In this work, we are concerned with hallucinations in code as a response to instruction prompts related to library usage. So, we required benchmarks with two key characteristics: (1) a natural language prompt that demands library usage and (2) test cases. Our evaluation focuses on Python because it is the most widely used language for coding tasks, and benchmarks with the stated characteristics are not available in other programming languages. Therefore, we evaluate with three open-domain code execution benchmarks: DS-1000, which has 1,000 problems over 7 libraries (Lai et al., 2023), Open-Domain EXecution-based natural language (ODEX), which has 945 problems in four natural

languages (en, es, ja, ru) over 79 libraries (Wang et al., 2023) and BigCodeBench (instruct), which has 1,140 problems over 139 libraries (Zhuo et al., 2025). In all of them, answers involve multiple libraries, either explicitly specified in the request or implicitly needed. Rather than using grammars that cover all libraries for every question, we use the ones needed: explicitly indicated ones and common related libraries.

**Metrics**  For detection, we consider precision and recall on (1) all errors, and (2) hallucination specific errors, which we call *Imaginary Features (IF)*. We quantify *Imaginary Features Recall* ($IF_R$) as:

$$IF_R = \frac{TP_{\mathcal{A}_{\mathrm{IF}}}}{TP_{\mathcal{A}_{\mathrm{IF}}} + FN_{\mathcal{A}_{\mathrm{IF}}}}$$

Where $\mathcal{A}_{\mathrm{IF}}$ is the set of *Imaginary Features (IF)*. Interchanging $\mathcal{A}_{\mathrm{IF}}$ for $\mathcal{A}_{\mathrm{E}}$ will be *Recall (R)* in the set of all errors.

Note that, for repair on Section 5.2, we define *Repair on Imaginary Features* ($R_{IF}$) as the ratio of how many imaginary features remain after repair, explained by Equation:

$$R_{IF} = \frac{\mathcal{A}_{\mathrm{IF}}}{\mathcal{A}_{\mathrm{IF}} + \mathcal{A}_{\mathrm{IF}} \backslash \mathcal{A}_{\mathrm{E}}}$$

Where $\mathcal{A}_{\mathrm{IF}}$ is the set of *IF* after repair, and the denominator is all errors after repair (the sum of $\mathcal{A}_{\mathrm{IF}}$ and the remaining errors $\mathcal{A}_{\mathrm{IF}} \backslash \mathcal{A}_{\mathrm{E}}$).

For mitigation, we use three metrics. *Pass@k (P@k)* evaluates functional correctness and returns the ratio of k samples passing a set of test cases (Haque et al., 2023)[2]. *Execution Rate (ER)* measures how frequently code executes, without considering correctness. $IF_\%$, which is the percentage of errors among all samples that are one of four types: attribute, import, type, and module not found. For examples of these errors, and which token causes the issue, see Table 7 in the Appendix. The equation of $IF_\%$ is described as:

$$IF_\% = \frac{\mathcal{A}_{\mathrm{IF}}}{\mathcal{A}}$$

Where $\mathcal{A}$ represents the set of all the samples in the benchmark.

**Generation Models**  We test our approaches on the output of six LLMs with even number of open and closed models: Claude-3, GPT-4, and GPT-3.5, Gemma2 2B, IBM-Granite 3B and Qwen 3 8B. For the API based models, we used a temperature of one, and for the open source model, we used 0.4. These values are as reported in prior work using these models on these benchmarks. The three models chosen are fine-tuned or have code in its training data. For example, the IBM Granite model is fine-tuned for tasks such as code fixing and explanation (Mishra et al., 2024). We validated samples for all 6 models on all questions in DS-1000 and BigCodeBench, and 550 questions from Odex (not the full dataset because we skipped questions that do not require library imports).

## 5.1  Detection

On detection, we see two distinct patterns of results. Table 1 presents precision and recall for our methods and the baseline on all errors. First, across all error types, the baseline, o3-mini, has high precision and low recall. Mypy and Pyright, the off-the-shelf static analysis tools, have very similar results, with higher precision than recall on DS-1000 and BigCode; however, on Odex SOTA models have higher recall than precision, as further discussed in Section 5.1.1.

Second, on Figure 1, all methods achieved a precision of 1 for the detected imaginary features, except for IBM-Granite on DS-1000, which achieved 0.98 precision. In particular, the grammar has high precision on the subset of imaginary features it targets but low recall, ranging from 6% to 20%. While grammars are commonly used to fix syntax bugs, we noticed that SOTA models occasionally generate these errors with a minimum of one occurrence (0.01%) and a maximum of 15.6%. The highest number of cases arises from the interchangeable generation of code and natural-language explanations, reinforcing the need to examine library-related hallucinations.

---

[2]We evaluate using Pass@1; so the LLM has a single run to give the correct answer.

Table 1: Detection results for all query models, detection methods, and datasets. We defined $P$ as precision and $R$ as recall when detecting all types of errors. Baseline in gray.

| | SOTA Models | | | | | | Small Open-source Models | | | | | |
| | claude-3 | | gpt-3.5 | | gpt-4 | | Gemma2 2B | | Granite 3B | | Qwen3 8B | |
| | P | R | P | R | P | R | P | R | P | R | P | R |
|---|---|---|---|---|---|---|---|---|---|---|---|---|
| **DS-1000** | | | | | | | | | | | | |
| o3-mini | **0.63** | 0.43 | **0.61** | 0.45 | **0.55** | 0.43 | **0.95** | **0.73** | 0.79 | 0.51 | **0.70** | 0.35 |
| Grammar | 0.36 | 0.14 | 0.45 | 0.11 | 0.53 | 0.15 | 0.54 | 0.11 | 0.57 | 0.10 | 0.34 | 0.12 |
| Mypy | 0.22 | 0.27 | 0.26 | 0.34 | 0.16 | 0.23 | 0.65 | 0.46 | 0.63 | 0.26 | 0.34 | 0.39 |
| Pyright | 0.61 | **0.70** | 0.57 | **0.60** | 0.50 | **0.62** | 0.85 | 0.58 | **0.95** | **0.55** | 0.60 | **0.59** |
| **Odex** | | | | | | | | | | | | |
| o3-mini | 0.61 | 0.09 | **0.61** | 0.22 | **0.79** | 0.32 | **0.79** | **0.30** | 0.74 | **0.34** | 0.65 | 0.06 |
| Grammar | 0.35 | 0.08 | 0.43 | 0.08 | 0.52 | 0.11 | 0.61 | 0.08 | 0.59 | 0.15 | 0.38 | 0.15 |
| Mypy | **0.85** | 0.15 | 0.39 | 0.66 | 0.43 | 0.81 | **0.79** | 0.04 | **0.81** | 0.18 | **0.80** | 0.07 |
| Pyright | 0.76 | **0.22** | 0.39 | **0.68** | 0.43 | **0.82** | 0.73 | **0.30** | 0.72 | **0.34** | 0.50 | **0.27** |
| **BigCode** | | | | | | | | | | | | |
| o3-mini | **0.64** | 0.12 | **0.56** | 0.16 | **0.57** | 0.11 | **0.81** | 0.43 | **0.77** | 0.48 | **0.78** | 0.16 |
| Grammar | 0.19 | 0.04 | 0.24 | 0.12 | 0.23 | 0.06 | 0.44 | 0.22 | 0.45 | 0.09 | 0.40 | 0.08 |
| Mypy | 0.16 | 0.05 | 0.22 | 0.07 | 0.24 | 0.09 | 0.55 | 0.10 | 0.47 | 0.09 | 0.27 | 0.08 |
| Pyright | 0.32 | **0.20** | 0.28 | **0.16** | 0.34 | **0.20** | 0.70 | 0.41 | 0.62 | 0.30 | 0.34 | **0.20** |

The subset of detected hallucinations shows that SOTA models with the aid of type-inference tools, such as Pyright, prove to be more reliable than the LLM judgment. On the contrary, o3-mini can detect hallucination patterns on small models, but performance downgrades as model size increases. A summary of $IF_R$ by model, tool, and benchmark can be found in the Table 10 in the Appendix.

We can see that the performance of each tool by error type reflects differences in its underlying knowledge sources. Rule checkers such as mypy and grammar perform best on `ImportError` and `ModuleNotFound` since they rely on syntax rules and import registries that only require verifying functions in the library. Pyright is broader by incorporating type stubs and inference from typed codebases, which can help resolve type mutations, improving its effectiveness against `TypeError` and `AttributeError`. o3-mini performs poorly on the simpler case of distinguishing imaginary solves for imports; however, it can serve as an inference tool for type mutation. This shows that static analysis is a cheap and reliable deterministic tool for identifying a hallucinated library, where LLMs' prior knowledge is ungrounded.

### 5.1.1 Investigating Potential and Overcoming False Negatives

To identify the upper bound on performance of static methods, we manually inspected a sample of 200 failure cases in the subset of Imaginary Features per benchmark (600 in total), with an even sample size across four models (claude-3, gpt-3.5, gpt-4, and Granite 3B). We labeled them according to whether we believed a static analysis approach could feasibly catch them, and why or why not this is possible due to their root cause.

**Annotation reliability and validation** One third of the annotations were labeled by two domain annotators for two purposes: a) to validate reliability and b) to establish guidelines and a systematic rubric for annotation, which can be found as a supplemental material of this work. This resulted in an average Cohen's kappa of 0.7863 across the eight annotated labels. All open-closed labels exceeded 0.8 agreement, while the categorical labels ranged from 0.69 to 0.77. These results demonstrated sufficient annotator reliability to proceed with a single annotation for the remaining dataset. More details on inter-annotation agreement can be found in the Appendix D.1.

We consider four metrics: *Caught* is the percentage of cases that were flagged. *TP* (True Positives) is the percentage of cases where the hallucination was correctly identified for the right reason. *Overlap* is the

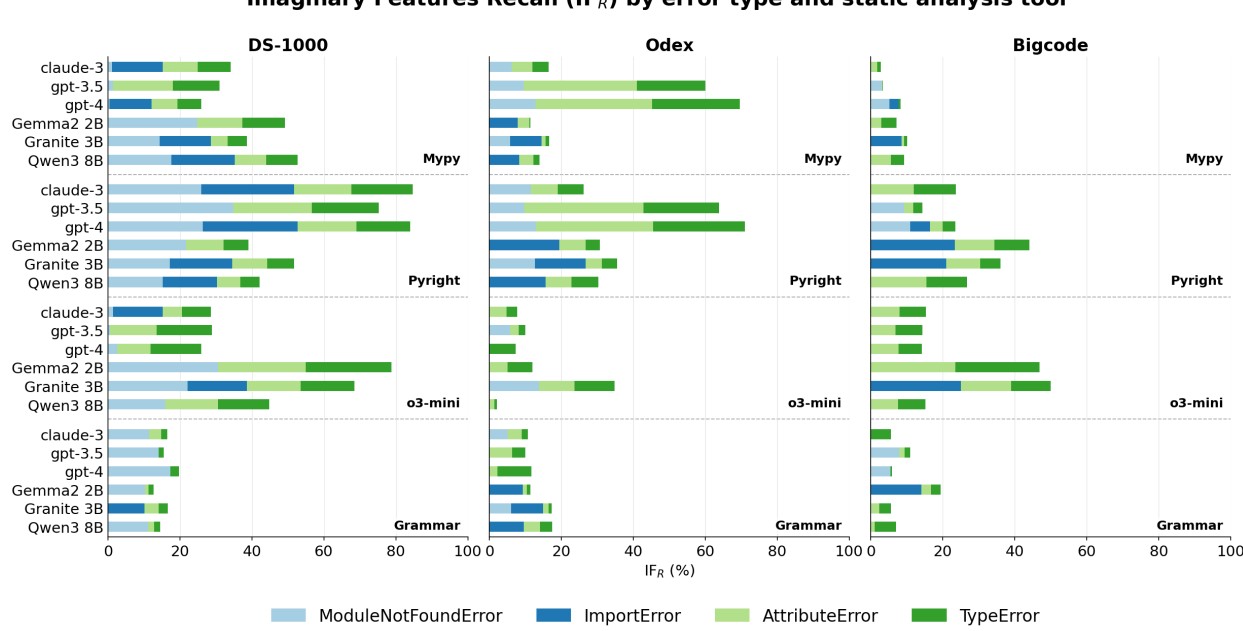

Figure 1: Detection results on $IF_R$ in percentage, which represents recall only on imaginary features. We do not show IF precision as it was 1 in all cases, except for o3-mini in Granite on DS-1000, which was 0.98.

percentage of cases identified by one tool that were also identified by the other tool. *Capable* is the percentage of cases we believe the method could potentially catch (and for the correct reason rather than by chance).

In Table 2, for *Caught* and *TP* columns, we observe a high rate of False Positives (FP) when manually examining why the hallucination was generated. Due to the FP, we observe variability in the *Overlap* column between the detected samples. However, in Odex, this is when the grammar is far from static. This is particularly evident in the datetime library, where the way it is imported collides with the built-in and module definitions, resulting in higher *TN* for the grammar that translates into high precision and recall, whereas for Static it is a *FP*.

**Detection boundaries of static methods**   Note that the *Capable* column is our manual analysis of cases where we believe the general method has the potential to detect a hallucination, not necessarily to fix it. Our analysis identifies Data and Control Flow operations as the most feasible avenue for improving static detection coverage. In the first case, depending on the problem and library information, datatypes might mutate during the program and become false negatives when undetected by a static tool. For example, when given a dataframe as input and calculating the average of a single column, it returns a series. However, when there are two columns, a dataframe is returned. Therefore, static tools should account for these cases in their data flow. For control flow, the benchmarks use a function in their test pipeline that will be automatically run in the test suite later. Currently, false negatives involving nested lambda, map, or apply functions represent a concrete, addressable gap; resolving them would meaningfully increase detection in data science benchmarks like DS-1000, which is a standard for data science libraries using dataframes.

In Table 2, we see that on Odex, static tools are capable of detecting 70.5% of hallucinations; this also translates into higher recall observed in Table 1. This is because many cases of Odex do not return anything in the function, resulting in returning None being an easy true positive for the Static tools to identify. However, this is not the case for BigCode, with a more diverse source of hallucinations.

**Type annotations remain a challenge for static analysis**   Manual inspection reveals the limits of type annotations in static analysis tools for detecting errors. One key issue with the grammar approach is the limited information in docstrings. For example, docstrings do not define namespaces, and so `int32` in

Table 2: Manual analysis of a sample of hallucinations across benchmarks. *Static* is the union of a hallucination been flagged by either Mypy or Pyright.

|  | Caught | TP | Overlap | Capable |
|---|---|---|---|---|
| **DS-1000** | | | | |
| Static | 56.7 | 37.6 | 17.9 | 77.0 |
| Grammar | 11.2 | 7.9 | 85.7 | 18.0 |
| **Odex** | | | | |
| Static | 19.0 | 4.0 | 12.5 | 70.5 |
| Grammar | 6.5 | 2.5 | 20.0 | 10.5 |
| **BigCode** | | | | |
| Static | 23.0 | 10.0 | 5.0 | 48.5 |
| Grammar | 8.5 | 5.0 | 100.0 | 7.0 |

Table 3: Categories that triggered a hallucination, along with their percentages on the sample of each benchmark.

| Hallucination cause | DS-1000 | Odex | BigCode |
|---|---|---|---|
| **During generation** | | | |
| Flow | 57.1 | 42.4 | 34.2 |
| Library | 14.7 | 6.6 | 6.0 |
| Data-Logic | 8.5 | 6.1 | 7.5 |
| Ambiguous input | 0.0 | 6.6 | 7.5 |
| **During test** | | | |
| Test case error | 2.8 | 4.0 | 6.5 |
| Ambiguous output | 1.1 | 30.3 | 26.6 |
| Test breadth | 15.8 | 4.0 | 11.6 |
| Prompt (Char count) | 871.8 | 87.5 | 663.2 |

`result = tf.random.uniform(..., dtype=tf.int32)` is marked as an error. However, this also occurs with more robust methods, such as Pyright and Mypy, where we observe instances where calls to functions are resolved with type stubs containing kwargs as a parameter in their annotations, for example, on `date_range = pd.date_range(start=start, end=end, format='%Y%m%d')`, static analysis tools are unable to detect `format` as an imaginary feature. With inaccurate and unspecified data types and dimensions in data structures, those hallucinations will become false negatives.

Another false negative for that grammar happens when it is unable to link an attribute to the library name or alias. To see the issue this causes, consider `cv = CountVectorizer(stop_words='english', punctuation_pattern=r'\\W')`. `punctuation_pattern` is a hallucination, but the grammar does not identify it because this code doesn't use the library name or alias in the call, instead importing the object using `from ... import CountVectorizer`. We could modify the grammar to avoid requiring the explicit use of the library name or alias, but that would introduce additional false positives. Tracking import statements would require features beyond those supported by llama.cpp.

### 5.1.2 Causes of Hallucinations and Identifying False Positives

For each hallucination in the sample, we also annotated the cause and whether it occurred during the LLM's code or while processing the test. These results yield allows us to distinguish between hallucinations that static methods could in principle resolve, and those that require fundamentally different approaches. Table 9 in the Appendix describes examples of root cause failures and why these could be detected based on the capability of each detection tool.

There were three hallucination categories that fall outside the scope of static analysis by nature, not by limitation of current tools, these relate to prompt underspecification rather than code analysis. (1) When a calculation requires the input data to design the code, e.g., a column with mixed data types. (2) When the request does not specify the input or output data type, or the prompt is ambiguous about the library it uses. (3) Execution errors that occur in testing code and are not part of the generated code.

Table 3 presents categories of the source of each hallucination in the sample of each benchmark in percentage. We consider hallucinations that are feasible to detect as those related to Control Flow and Data Flow (*Flow*) in static tools, as well as rule definitions that match the *Library* definitions. These two represent the majority of cases in DS-1000. However, for Odex and BigCode, we have a more diverse distribution. For type (1), hallucinations that depend directly on the knowledge of the LLM, such as setting a variable to a negative value without reason and then encountering an exception because it requires positive numbers, are definitely outside the scope of the tools.

Table 4: Code performance with various repair methods. The percentage of responses that are executable (ER) and the ones that are correct (P@1). Baseline in gray.

| | SOTA Models | | | | | | Small Open-source Models | | | | | |
| | claude-3 | | gpt-3.5 | | gpt-4 | | Gemma2 2B | | Granite 3B | | Qwen3 8B | |
| | ER | P@1 | ER | P@1 | ER | P@1 | ER | P@1 | ER | P@1 | ER | P@1 |
|---|---|---|---|---|---|---|---|---|---|---|---|---|
| **DS-1000** | | | | | | | | | | | | |
| No repair | 73.1 | 42.7 | 73.2 | 36.6 | 76.8 | 47.7 | 47.2 | 5.9 | 33.6 | 8.9 | 70.1 | 34.2 |
| o3+Self | **82.3** | **44.3** | **82.8** | **37.9** | **85.3** | **49.3** | **77.9** | **9.7** | **56.3** | **12.0** | **84.3** | **36.1** |
| o3+Mypy | 80.0 | 43.7 | 80.3 | 37.7 | 82.5 | 49.1 | 59.3 | 7.9 | 45.5 | 10.1 | 75.6 | 34.8 |
| o3+Pyright | 80.3 | **44.3** | 79.9 | 37.4 | 82.1 | 48.7 | 57.8 | 7.6 | 44.5 | 10.1 | 74.4 | 34.8 |
| **Odex** | | | | | | | | | | | | |
| No repair | 66.9 | 42.2 | 59.6 | 35.8 | 57.0 | 35.2 | 49.4 | 23.5 | 46.3 | 18.0 | 57.1 | 33.2 |
| o3+Self | 70.3 | 43.6 | 67.3 | 40.4 | 67.5 | **43.2** | **69.4** | **34.6** | 50.3 | **19.6** | **68.0** | **39.5** |
| o3+Mypy | 71.1 | **44.2** | 67.3 | 40.2 | 67.5 | 43.0 | 51.6 | 24.7 | 48.5 | 19.0 | 60.7 | 35.0 |
| o3+Pyright | **71.3** | 44.0 | **67.7** | **40.6** | **68.1** | **43.2** | 56.1 | 28.0 | **50.3** | 19.4 | 61.4 | 35.8 |
| **BigCode** | | | | | | | | | | | | |
| No repair | 79.0 | 45.5 | 77.6 | 39.3 | 79.6 | 46.0 | 50.6 | 10.0 | 61.0 | 20.6 | 76.0 | 40.9 |
| o3+Self | **82.6** | **47.3** | **82.1** | **41.3** | **82.6** | **47.2** | **77.5** | 20.0 | 77.7 | 26.7 | **83.3** | **44.9** |
| o3+Mypy | 79.0 | 45.5 | 77.9 | 39.4 | 80.0 | 46.1 | 53.2 | 10.9 | 62.2 | 21.1 | 76.7 | 41.1 |
| o3+Pyright | 80.0 | 45.9 | 78.3 | 39.4 | 80.4 | 46.1 | 61.7 | 13.9 | 66.7 | 23.2 | 77.8 | 41.5 |

**Blind spots in benchmarks' design** A non-trivial proportion of apparent hallucinations (over 30%) originate from benchmark underspecification rather than model failure, highlighting an important methodological consideration for future evaluation work. For type (2), an execution error may occur during the generated code if the prompt does not specify the type of input. A common case in BigCode is using the keyword 'Dataset'. The LLM designs code for a dataframe, but the function is meant to use a NumPy array as input and output. Another example is test cases containing None in the data. While some prompts specify the requirement to consider data that might contain None values, others do not, and have a test case that evaluates this. These two causes in DS-1000 represented 1.1%, while in Odex 36.9%, and in BigCode (instruct) 34.1% over the annotated sample. We associate these with the prompt character count; DS-1000 has almost 800 more characters than Odex and almost 200 more characters than BigCodeBench.

For type (3), we observed that the three benchmarks contain test cases that fail during the initialisation of input data. We labeled these as *Test case error* and found that they appear 2.8-6.5% of the time. Finally, with benchmarks that test the breadth and depth of library function calls, we consider *Test breadth* as whether the assertion cases have different ways to query the library from a valid solution. An example is when asking to set a title in a plot; there are several ways to evaluate this. One could use either `ax.get_title()` or `plt.gca().get_title()` depending on how the LLM sets the title; however, these three benchmarks only consider one of those options. In reality, we need to consider different valid options in the library to account for diverse valid solutions.

## 5.2 Repair

In this section, we turn from detecting errors to repairing them. We evaluate on the examples that contained an error when executed: 2210 samples in DS-1000, 2001 in BigCode, and 1291 in Odex (across all six generation models). To repair the error, we prompt an LLM with the code, the error, and a request to resolve the issue. We try o3-mini combined with static analysis tools, or no tool at all. We average over three runs.

Table 4 shows that our approach consistently improved all metrics, particularly execution rate (ER). Meanwhile, Table 5 shows *Repair on Imaginary Features* ($R_{IF}$), which addresses when repair fails, how often it is due to an imaginary feature? In most cases, there is no improvement in fixing hallucinations; moreover, in Odex, the hallucination rate increased. Interestingly, o3-mini performs better without static analysis in-

Table 5: Code performance with various repair methods. $R_{IF}$ is the rate of imaginary features (IF) relative to all remaining errors after repair; lower is better. Baseline in gray.

| | SOTA Models | | | Small Open-source Models | | |
| | claude-3 | gpt-3.5 | gpt-4 | Gemma2 2B | Granite 3B | Qwen3 8B |
| | $R_{IF}$ | $R_{IF}$ | $R_{IF}$ | $R_{IF}$ | $R_{IF}$ | $R_{IF}$ |
|---|---|---|---|---|---|---|
| **DS-1000** | | | | | | |
| No repair | 9.1 | 9.7 | 8.1 | 22.3 | 17.2 | 25.4 |
| o3+Self | **2.7** | **2.6** | 1.9 | 8.1 | **11.9** | 13.4 |
| o3+Mypy | 2.9 | 3.3 | **1.7** | **5.8** | 12.7 | 6.5 |
| o3+Pyright | 3.2 | 3.5 | 2.4 | 6.4 | 13.9 | **6.1** |
| **Odex** | | | | | | |
| No repair | 20.4 | **15.8** | **13.7** | **23.8** | **24.0** | **31.3** |
| o3+Self | 16.6 | **15.8** | 16.0 | 51.0 | 46.6 | 47.8 |
| o3+Mypy | **16.2** | 16.8 | 15.6 | 27.8 | 48.7 | 55.6 |
| o3+Pyright | **16.2** | 16.4 | 16.0 | 45.2 | 48.9 | 51.4 |
| **BigCode** | | | | | | |
| No repair | 30.1 | 35.7 | 36.6 | 44.9 | 43.9 | 31.8 |
| o3+Self | 29.8 | **34.3** | 35.4 | 42.7 | 39.0 | 28.9 |
| o3+Mypy | 30.5 | 35.7 | 36.4 | **35.7** | 39.0 | **23.1** |
| o3+Pyright | **28.9** | 36.4 | **34.5** | 40.8 | **38.2** | 36.4 |

formation on DS-1000 and BigCodeBench, but not on Odex. Notably, repair improves overall executability consistently, but imaginary features prove resistant, a finding that reveals detection and repair as fundamentally decoupled challenge.

### 5.3 Mitigation

Finally, we consider avoiding generation of errors entirely by using constrained decoding. This can only be done with the grammar approach and an open-source model. Table 6 shows that constrained decoding reduces imaginary library features ($IF_\%$) in Odex, where repair alone could not; however, results on DS-1000 and BigCodeBench [3] vary by model. The modest shifts in ER and P@1 are expected given the grammar's targeted scope, and confirm that constrained decoding acts as a precise, low-overhead intervention.

**Why did *Pass@1* decrease?** Pass@1 decreases in some cases because grammar constraints derived from docstrings reflect the completeness of library documentation, an important consideration for future grammar construction. This is consistent with work on extracting hyperparameter constraints from machine learning operators, which found low precision in docstrings and proposed other methods like weakest-precondition (Rak-Amnouykit et al., 2021). Another case is when valid alternative solutions fall outside the LLM's sampling distribution; even correct candidates in the grammar might not appear in the LLM's pretraining data. Figure 2 shows an example of how the grammar can help. Without constrained decoding, the LLM produced an imaginary parameter `use_line_collection`; with constrained decoding, it generates valid parameters but not the correct answer, suggesting the issue is distributional rather than grammatical. Recent work on hallucination benchmarks (Ravichander et al., 2025) suggests that coding tasks, such as library hallucinations, appear in pretraining data examples. Code that may be right in a specific document, in isolation, may be incorrect when used later.

Despite the additional resampling overhead, constrained decoding remains computationally practical. We observed that an unconstrained response takes an average of 22.3 seconds, while a constrained one takes an average of 34.76 seconds, adding an average of 12 seconds per request, which varies with resampling frequency. Therefore, we evaluated *Sampling cost* (Olausson et al., 2023), the total number of tokens sampled from the model. Odex averaged 1.56 resamplings per request (one to three additional samplings) versus 3.32 for

---

[3]In Table 4 we used IBM-Granite's responses from the BigCodeBench Leaderboard; however, there was no specification of the IBM-Granite version, so we re-run IBM-Granite with our version to do a fair comparison with our constrained model.

Table 6: Benchmark results with and without constrained generation using our grammar. Imaginary Features ($IF_\%$) as the percentage of hallucinations over the benchmarks' sample. Baseline in gray.

| | Gemma2 2B | | | Granite 3B | | | Qwen3 8B | | |
|---|---|---|---|---|---|---|---|---|---|
| | ER | P@1 | $IF_\%$ | ER | P@1 | $IF_\%$ | ER | P@1 | $IF_\%$ |
| **DS-1000** | | | | | | | | | |
| Unconstrained | 47.2 | 5.9 | **10.4** | 33.6 | 8.9 | 10.4 | 70.1 | 34.2 | 6.7 |
| Constrained | 46.2 | 6.3 | 12.6 | 36.3 | 6.5 | **6.7** | 61.5 | 29.9 | **6.6** |
| **Odex** | | | | | | | | | |
| Unconstrained | 49.4 | 23.5 | 5.4 | 46.3 | 18 | 8.9 | 57.1 | 33.2 | 6 |
| Constrained | 41.7 | 20.8 | **4.8** | 45.9 | 18.6 | **8.2** | 50 | 28.3 | **4.4** |
| **BigCode** | | | | | | | | | |
| Unconstrained | 50.6 | 10 | 22.2 | 58.1 | 16 | **18.4** | 76 | 40.9 | **7.6** |
| Constrained | 56.3 | 11.5 | **18.1** | 54.2 | 12.7 | 21.2 | 75.4 | 38.5 | 8.2 |

```
# make a stem plot of y over x
# and set the orientation to be horizontal
plt.stem(x, y, use_line_collection=True)
plt.show()
```

```
# make a stem plot of y over x
# and set the orientation to be horizontal
plt.stem(x, y, linefmt='C0-', markerfmt='o',
basefmt='C0-')
plt.xlabel('x')
plt.ylabel('y=e^{sin(x)}')
```

Figure 2: Example of how constrained decoding impacts output. **On the top:** Unconstrained response with imaginary feature. **On the bottom:** Constrained response with factual parameters.

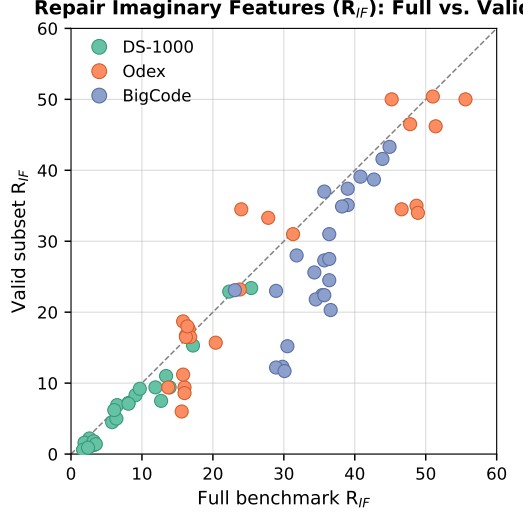

Figure 3: Repair ($R_{IF}$): Full vs. Valid subset. Points represent a combination of model and repair condition across benchmarks.

DS-1000 (three to eight additional samplings). On Odex, the 12-second overhead reduces hallucinations by 0.7 points in $IF_\%$; doubling the overhead in DS-1000 yields a larger reduction of 3.7 points. This could be explained by the distribution of rules in each grammar, since in DS-1000, we found more uniform and larger grammar rules, as seen in Figure 8 in the Appendix.

### 5.4 Validity Against Benchmark Noise

We run the analysis on imaginary features (IF) on the three experiments for the subset of questions identified as valid. This means we removed the questions where the requested input and output were unclear, and the test suite provided contains coding errors as identified in Section 5.1.2. This translated into removing 3% of DS-1000, 7.7% in Odex, and 6.1% in BigCodeBench. Then, we calculated the change in the full set minus the validated set for detection ($\Delta IF_R$), repair ($\Delta R_{IF}$), and mitigation ($\Delta IF_\%$).

We conducted a Spearman correlation for each experiment and benchmark between the original metric and the valid subset metric, and found that the ranking of models' performance was significantly preserved at

0.001, indicating it was not affected by benchmark noise. For detection, the average correlation was 0.98 with average $\Delta IF_R = 0.02$, and for constrained decoding, the average correlation was 0.99 with average $\Delta IF_\% = 0.60$; Figure 12 in the Appendix shows an almost perfect correlation between these two metrics, while in percentage removing the subset will change the results by less than 1%. In repair, we have a different case with a lower average p=0.88, significant at 0.001. As seen in Figure 3, across the three benchmarks, the valid subset $R_{IF}$ is lower than the full set of $R_{IF}$, with an average $\Delta R_{IF} = -4.1$, suggesting that approximately 4% of repair failures cannot be fixed due to the benchmark rather than the model.

## 6 Discussion

This work establishes an upper bound on the performance of static analysis for library hallucination detection. While automated detection appears to reach up to 85% of hallucinations, after manual inspection we found that false positives mean the true upper bound is 77%. This gap is explained by false positives arising from limitations in how static tools handle dynamic types in Python. Quantifying this ceiling is important for the users to understand the tools' capabilities, and suggests that detecting library hallucinations requires approaches that go beyond static analysers.

Detecting and repairing hallucinations are two separate avenues. Section 5.2 shows that although repair improved execution and Pass@1, the hallucination rate remained unchanged across all benchmarks, while constrained decoding reduced hallucination only in Odex. This is consistent with findings that grammar-constrained decoding can distort the model's probability distribution, producing grammatically valid but semantically incorrect tokens (Park et al., 2024); similarly, repair guided by static analysis does not shift the model's underlying distribution toward correct library features, leaving hallucinated tokens as the most probable candidates.

Unlike approaches that sample higher-quality tokens without explicitly addressing detection and mitigation simultaneously, this work provides a framework that directly targets both, though fully eliminating hallucinations remains an open challenge for the field (Tanzil et al., 2024). However, as shown in Figure 1, static analysis tools and grammars are useful for detecting hallucinated packages before execution and can prevent malicious packages from running on users' systems (Spracklen et al., 2025; Krishna et al., 2025).

Beyond mitigation, such detection tools may also serve as evaluation instruments for comparing LLMs' tendencies to hallucinate library features, which can potentially influence the development of code models. Together, our findings suggest that future directions for this problem may benefit from methods beyond the output-level interventions tested in this study, static analysis, and grammar-constrained decoding, and more towards understanding how these hallucinations originate in the first place.

## 7 Conclusions

This work provides the first empirical analysis of static analysis and grammar-constrained decoding methods for detecting and mitigating library hallucinations in NL-to-code settings. Our evaluation is limited to Python, a choice made because it is the dominant language for execution-based benchmarks. While the framework is general, the grammar construction method for other languages would require non-trivial adaptation, as it relies on the languages' formal grammars and on library inspection. Within this scope, our manual analysis establishes a principled upper bound of 77% for static hallucination detection and reveals the blind spots that emerge: type inference and control-flow tracking. While the biggest strength of static tools is detection, our results show that this does not transfer to repair or mitigation. Finally, we suggest that work on detecting code hallucinations will need to employ different methods; one possible avenue is the model's internal state.

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

## A  Appendix

## B  Grammar Creation

Our grammar is in GBNF syntax, which can be directly used with *llama.cpp* to constrain decoding.

Our two-step approach provides modularity that could make adaptation to other programming languages easier in the future.

Almost all of this processing is automatic. The Python language formal grammar in EBNF form was manually converted into GBNF in 4 hours of manual work. The library grammars were automatically extracted. Common aliases for libraries (e.g., np for numpy) were automatically extracted from use in the benchmark with a regular expression. As shown in the Appendix 6, each step takes less than a second, as indicated by the distribution across all the libraries tested, demonstrating the viability of this approach for integration at a low cost into engineering pipelines. To expand to more libraries, defining common aliases is the only step that requires validation. One limitation of our approach is that it cannot handle atypical aliases because llama.cpp does not allow the grammar to track state. However, atypical aliases are rare and likely to become even more rare as LLM use rises and so the same alias is consistently proposed.

Figure 4 shows a slightly simplified snippet of Numpy's grammar. We highlight in bold the main symbols that verify a program with the grammar. Starting from the symbol **root**, we can accept all tokens that describe instructions in natural language, except for the prefix of the library name *numpy* or alias *np*. If a library prefix is matched, all the following tokens should also match those from the **library** symbol, e.g., numpy methods, builtins, or constants. Similarly, matching the token *import* will add as next states the **import-from** and **import-name** symbols, forcing the verification of subsequent tokens as Python's builtins and environment modules.

We conducted our experiments on a local computer with an M2 processor, 8-core CPU, 10-core GPU, and 24GB RAM.

## C  Data

We used BigCodeBench under the Apache 2.0 license, and DS-1000 and ODEX under cc-by-sa-4.0. We use those for evaluation, and we did not modify their original content.

## D  Imaginary features

In Table 7, we present an analysis of the generated code and highlight the tokens that we consider to be imagined by an LLM in the library usage context, and that will result in an error when the code is executed.

```
# Connector rules between the library and the GBNF
root ::= ( except-library | library | import )+
except-library ::= ( "numpy" [^.] | "np" [^.] | "from" [^ ] | "import" [^ ] )
library ::= ( "numpy." | "np." ) ( np-numpy-methods | np-numpy-builts | np-numpy-const )
class-name ::= ( "n"[^p] | "nump"[^y] )
class ::= class-name (trailer)*
# Import statements
import ::= ( import-name | import-from ) "\n"
import-from ::= "from " dotted-name " import " name ( " as " name)?
import-name ::= "import " dotted-name (" as" name)?
dotted-name ::= ( python-modules | environment-modules )
# Built-ins and constant rules
np-numpy-builts ::= "char" | "compat" | "compat.py3k" | "compat.tests" | "core" ...
np-numpy-const ::= "e" | "euler_gamma" | "inf" | "nan" ...
np-numpy-methods ::= np-numpy-copyto | ...
```

Figure 4: Python's partial GBNF with a subset of rules defining the *numpy* library.

Table 7: Examples and descriptions of type of bugs considered as imaginary features. The tokens considered as imaginary are highlighted in red.

| Imaginary Features | Description |
|---|---|
| **TypeError** | A common example is a call to a function that does not match the function's definition, such as parameters with incorrect datatype or name. |
| Generated code | **pd.DataFrame(df, columns=df.columns, suffixes=('_d', '_z'))** |
| **AttributeError** | This occurs when trying to access an attribute that does not belong to the class, module or submodule. |
| Generated code | C = tf.tensordot(A, B, axes=[[2], [2]]) 
 sess = **tf.Session()** 
 print(sess.run(C)) |
| **ImportError** | This happens when trying to solve a built-in class or function from a library, and it is not a member. |
| Generated code | **from sklearn.externals import joblib** 
 joblib.dump(fitted_model, 'sklearn_model.pkl') |
| **ModuleNotFoundError** | This module does not exist in the environment and/or in real life. |
| Generated code | **import numpy_indexed as npi** 
 result = npi.group_by(accmap).sum(a) |

## D.1 Annotation

We provided annotators with an annotation guideline containing five sections. This guideline is available as supplemental material.

- Context: here we define what a hallucination is in code and educate annotators about the relevance of each hallucination in a coding environment, using an example.

Table 8: Cohen's kappa analysis of inter-annotator agreement between 25% of annotated labels.

| Label | N | % Agreement | Cohen's Kappa | Interpretation |
|---|---|---|---|---|
| **Open-close (Yes / No / NA)** | | | | |
| Caught Static | 98 | 94.9 | 0.8849 | Almost Perfect |
| Capable Static | 163 | 89.57 | 0.7181 | Substantial |
| Caught Grammar | 17 | 94.12 | 0.8496 | Almost Perfect |
| Capable Grammar | 163 | 92.02 | 0.767 | Substantial |
| **Categorical (capabilities / NA)** | | | | |
| Caught Static Reason | 104 | 88.46 | 0.8422 | Almost Perfect |
| Capable Static Reason | 163 | 77.91 | 0.7315 | Substantial |
| Caught Grammar Reason | 17 | 88.24 | 0.8111 | Almost Perfect |
| Capable Grammar Reason | 163 | 77.3 | 0.6857 | Substantial |
| **Average** | | | **0.7863** | **Substantial** |

- Data: we describe each column that they need to annotate and summarize the expected output for each column.

- Tools: we provide an overview of each tool's capabilities and limitations (static analyzer and grammar), and we highlight examples of difficult cases.

- Examples: we provided 16 pages of examples, each containing the reasoning behind the annotation. Hard cases were run step by step, and the reasoning was provided in a Table.

Table 9 lists the labels in the categorical column for the reasons behind the tool's detection capability, and it summarizes the annotation guideline in the third section, "Tools".

Table 8 reports the Cohen's Kappa agreement for each labeled column. N (valid pairs), which includes only cases where the tool detected something; otherwise, they have the NA "nan" value. As you can see, caught grammar has fewer pairs, since it is the tool with fewer detected hallucinations. When the "caught static reason" column is higher than the "caught static" column, it indicates that an annotator thought a "test error (that occurred in test)" did not apply to the tool's detection; this was later clarified in the instructions and guidelines.

### D.2 Detect

In Figure 5, we show an example of the prompt we used for the LLM-as-a-judge baseline experiment. While the incorrect token is highlighted in red, we also show the LLM's response. Note that this example is an incorrect assertion made by an LLM. In Figure 7, we show the distribution of the time taken on each step in the grammar construction. In Figure 6, we compare the time taken for each method to detect a bug. *Mypy* was the fastest detection tool with a mean of 1s, followed by *Pyright* and grammar with an average of 2.5s, and o3-mini with a mean of 4s. In Table 2 10 we summarize the Imaginary Features Recall as displayed in Figure 1 by model, benchmark, and tool.

### D.3 Mitigation

Figure 8 shows the distribution of the number of rules used in each question between the benchmarks, and Figure 9 compares the distribution of resample tokens for each benchmark. Figure 10 shows the distribution of the time taken between the constrained and unconstrained versions of IBM-Granite in BigCodeBench.

Table 9: Examples and descriptions of annotation categories for the capability of tools to detect errors and their source.

| Static Analysers |
| --- |
| Can do the same as in the grammar (**Syntax + Library**) |
| Can keep data and control flow (**Flow**) |
| `cont = CountVectorizer()` |
| is of type CountVectorizer, and should have N attributes. |
| Should keep track of (**Lambda**) |
| Can know global and local scopes (**Scope**) |
| `def random(char)` vs `from random import random` |
| `def shift(something)` vs `shift =` |

| Grammars |
| --- |
| Find syntax errors (**Syntax**) |
| Closing parenthesis. |
| Variable names cannot start with digits. |
| A toke must be at the right of = |
| Functions and if statements should not be empty and must be followed by a statement block. |
| Docstring grammar (**Library**) |
| Defines the function in a library |
| Defines a parameter in the function |
| Defines a constant in the library |

| Infeasible |
| --- |
| **TEST**: the error occurs during the test suite, so an error message generated from a test case is not handled by these tools. |
| **AMBIGUOUS**: the tool needs to know the input and output descriptions of datatypes in the prompt to generate code accordingly. |
| **LOGIC-DATA**: the tool cannot know if filtering a dataframe will make the series empty, it just knows it is a dataframe. Or if the LLM output is too vague or has no code e.g. `END SOLUTION`. |

### D.3.1 Memoisation

As seen in Figure 11, we found no gains in time spent per token decoded. Further analysis is needed on the latency generated by Ollama and resampling. As for the remaining work on memory usage on pushdown states in both methods.

### D.3.2 Subset validity

Figure 12 shows the correlation between the full benchmark set and the valid benchmark subset identified by this work. We can see an almost perfect correlation.

Table 10: Detection results on $IF_R$, which represents recall only on imaginary features. We do not show IF precision as it was 1 in all cases, except for o3-mini in Granite on DS-1000, which was 0.98. Baseline in gray.

| | SOTA Models | | | | Small Open-source Models | | |
| | claude-3 | gpt-3.5 | gpt-4 | | Gemma2 2B | Granite 3B | Qwen3 8B |
| | $IF_R$ | $IF_R$ | $IF_R$ | | $IF_R$ | $IF_R$ | $IF_R$ |
|---|---|---|---|---|---|---|---|
| **DS-1000** | | | | | | | |
| o3-mini | 0.29 | 0.29 | 0.26 | | **0.78** | **0.68** | 0.45 |
| Grammar | 0.16 | 0.15 | 0.20 | | 0.13 | 0.17 | 0.15 |
| Mypy | 0.34 | 0.31 | 0.26 | | 0.49 | 0.36 | **0.53** |
| Pyright | **0.85** | **0.75** | **0.84** | | 0.39 | 0.52 | 0.42 |
| **Odex** | | | | | | | |
| o3-mini | 0.08 | 0.11 | 0.08 | | 0.13 | 0.34 | 0.02 |
| Grammar | 0.11 | 0.10 | 0.12 | | 0.08 | 0.17 | 0.18 |
| Mypy | 0.17 | 0.60 | 0.70 | | 0.12 | 0.17 | 0.13 |
| Pyright | **0.26** | **0.64** | **0.71** | | **0.31** | **0.36** | **0.30** |
| **BigCode** | | | | | | | |
| o3-mini | 0.15 | **0.14** | 0.14 | | **0.47** | **0.50** | 0.15 |
| Grammar | 0.06 | 0.11 | 0.06 | | 0.19 | 0.06 | 0.07 |
| Mypy | 0.03 | 0.03 | 0.08 | | 0.07 | 0.10 | 0.09 |
| Pyright | **0.24** | **0.14** | **0.24** | | 0.44 | 0.36 | **0.27** |

```
Prompt:
import numpy as np
import pandas as pd
import matplotlib.pyplot as plt

x = np.arange(10)
y = np.arange(10)
# Make two subplots. Make the first subplot three times wider
than the second subplot but they should have the same height.
# SOLUTION START
fig, (ax1, ax2) = plt.subplots(1, 2, figsize=(12, 4), gridspec_kw={'width_ratios': [3, 1]})
ax1.plot(x, y)
ax1.set_title('First␣Subplot')
ax2.plot(x, y**2)
ax2.set_title('Second␣Subplot')
plt.tight_layout()
plt.show()

Tell me if this code will execute, answer yes or no, followed by a one-line explanation.

LLM's response:
Yes - The code correctly creates two subplots with the first
being three times wider than the second.
```

Figure 5: Example prompt and response from LLM-as-judge to detect code that is executable.

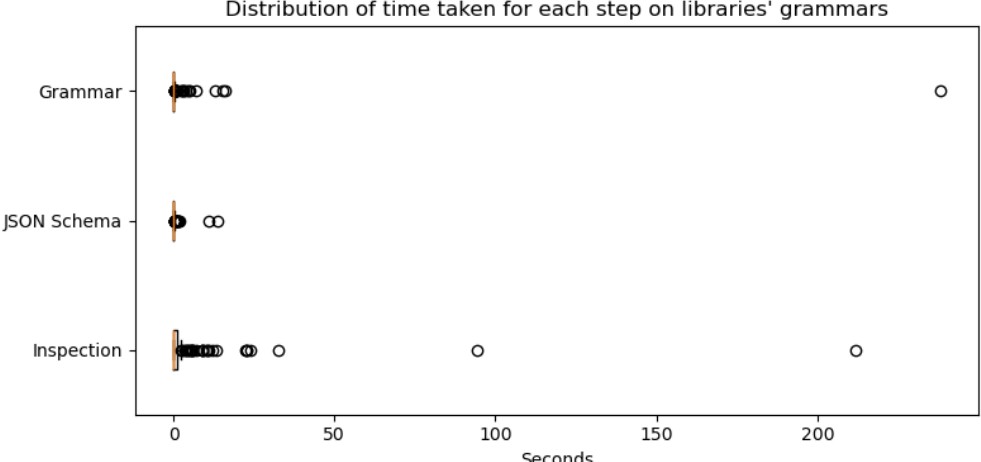

Figure 6: Distribution of time taken on each step to build the grammar.

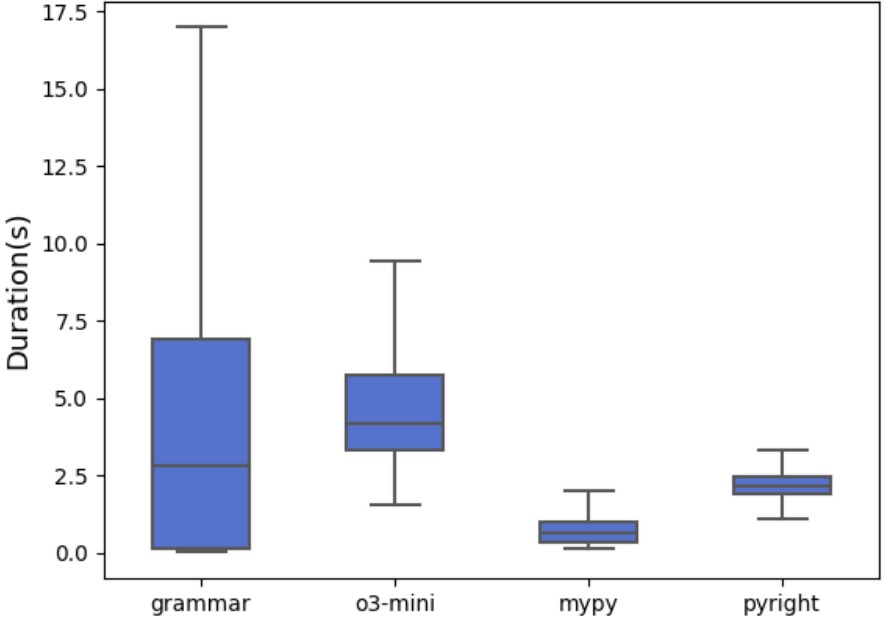

Figure 7: Comparison of time taken on bug detection analysis tools.

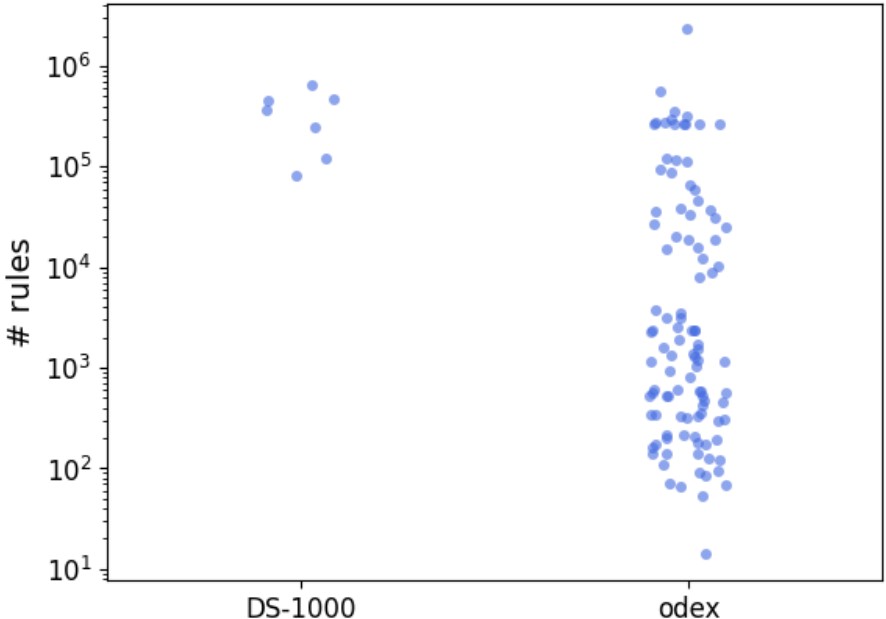

Figure 8: Distribution of the number of rules in the grammar for each benchmark.

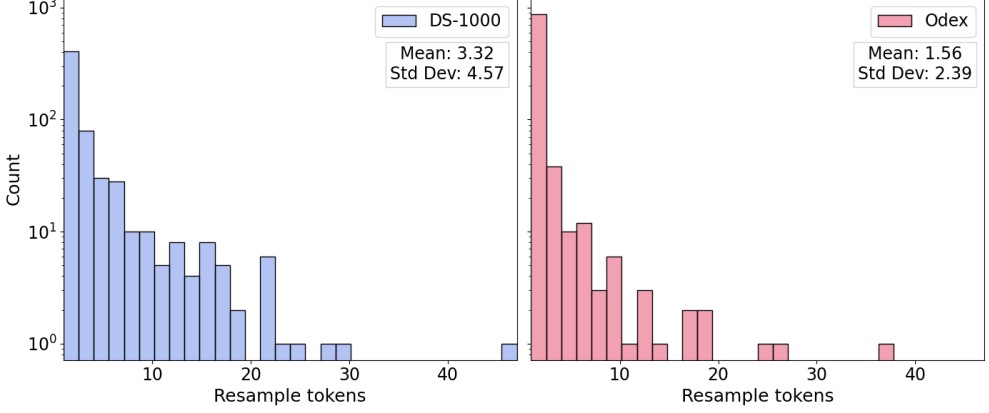

Figure 9: Distribution on resampling in each benchmark.

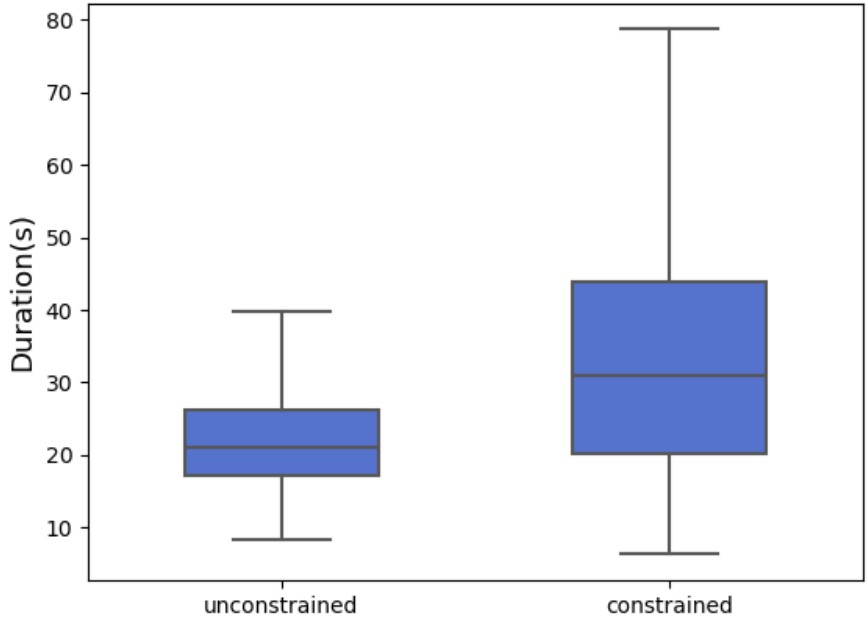

Figure 10: Comparison of time taken between unconstrained and constrained.

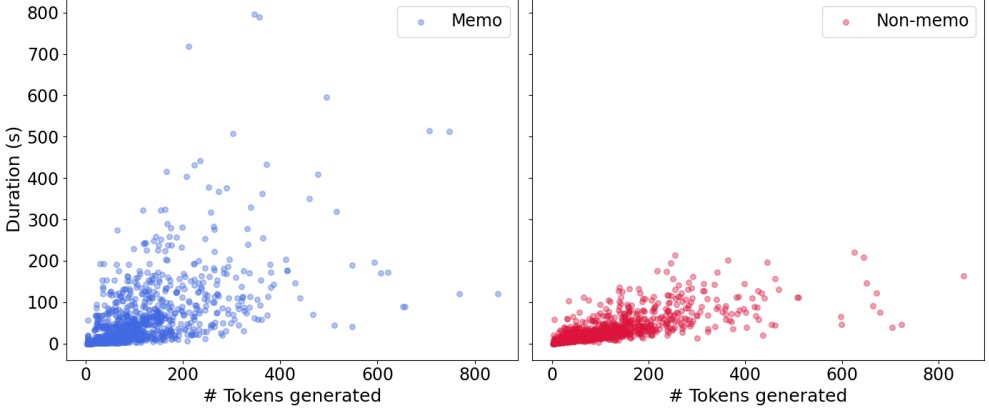

Figure 11: Comparison of time taken for approaches on grammar-constrained parsing.

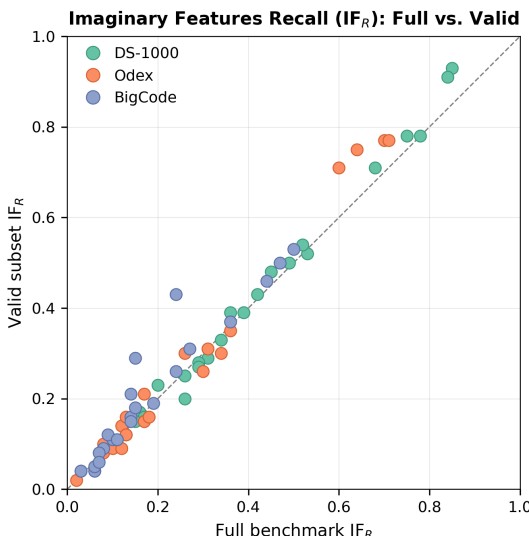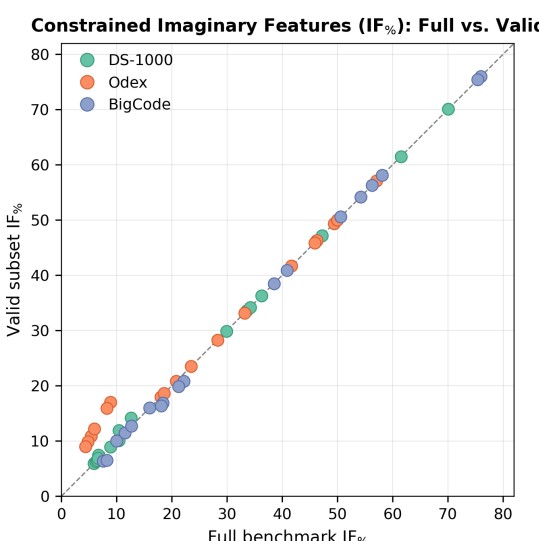

Figure 12: Validity against benchmark noise for detection and constrained decoding experiments. **On the left:** Imaginary Features Recall (IFR) full benchmark vs. valid subset. **On the right:** Constrained Imaginary Features (IF%) full benchmark vs. valid subset. Points represent a combination of model and repair condition across benchmarks.

