# OpenReview forum: "An Empirical Analysis of Static Analysis Methods for Detection and Mitigation of Code Library Hallucinations"
_TMLR — Under review for TMLR_

### Review · Reviewer_WGVv · 2026-06-02

**Summary Of Contributions:**

The paper presents an empirical study of whether static analysis methods can detect and mitigate code library hallucinations in LLM-generated Python code. It evaluates off-the-shelf tools such as Mypy and Pyright, an automatically constructed grammar-based method, constrained decoding, and an LLM-as-judge baseline across several existing NL-to-code benchmarks. A key contribution is the systematic quantification of what these methods can and cannot catch, supported by manual annotation of hallucination cases and an analysis of the upper bound of static detection. The main strength is that the paper provides a useful and practically motivated boundary analysis rather than simply proposing another repair method. The results also show that detection improvements do not reliably translate into repair or hallucination reduction.

**Audience:**

Yes

**Audience Explanation:**

Yes. TMLR’s scope includes empirical and analytical studies that provide insight into the behavior, evaluation, and reliability of learning systems. This paper fits that scope because it studies library hallucinations in LLM-generated code and evaluates whether static analysis, grammar-based checking, constrained decoding, and LLM-as-judge methods can detect or mitigate such errors. The paper does not primarily propose a new model, but it provides a systematic empirical analysis of an important reliability problem in code-generating LLMs. Its findings should be of interest to at least some TMLR readers, particularly those working on code LLMs, hallucination detection, program analysis for ML systems, and evaluation methodology.

**Broader Impact Concerns:**

The paper studies detection and mitigation of library hallucinations in LLM-generated code, which is generally beneficial for improving the reliability and safety of code generation systems. One minor point is that the paper could more explicitly discuss the security implications of package or API hallucinations, since hallucinated library names or imports may create opportunities for dependency-confusion or malicious-package attacks.

**Claims And Evidence:**

Yes

**Claims Explanation:**

* **Claim: Static analysis can detect a subset of library hallucinations.**
  The experimental design supports this claim. The paper evaluates multiple detection methods, including Grammar, Mypy, Pyright, and an o3-mini-based judge, across DS-1000, ODEX, and BigCodeBench. It reports detection precision and recall, and further distinguishes between overall errors and imaginary features, which directly matches the paper’s focus on library hallucinations.

* **Claim: Static analysis has clear limitations.**
  This claim is also well supported. The authors conduct a manual analysis to determine which hallucination cases are plausibly detectable by static methods and which are not. The estimated upper bound of 48.5%–77% provides direct evidence for the paper’s boundary-analysis goal.

* **Claim: Repair/mitigation is not equivalent to detection.**
  The experiments support this distinction. Repair improves executability and Pass@1, but does not consistently reduce the remaining imaginary features. Similarly, grammar-constrained decoding is only effective in some settings. The Discussion and Conclusion appropriately emphasize that the main strength of static tools lies in detection, and that this does not automatically transfer to repair or mitigation.

* **Claim: Benchmark ambiguity is itself an important issue.**
  The paper treats benchmark ambiguity as an empirical finding rather than a confound to be removed. This is reasonable given the paper’s stated goal. The authors show that some apparent hallucinations are caused by prompt underspecification, ambiguous outputs, limited test breadth, or test-case artifacts. This analysis is consistent with the paper’s positioning as an empirical study of static analysis on existing NL-to-code benchmarks.

**Requested Changes:**

The paper is generally well aligned with its stated goal as an empirical analysis of static analysis methods for library hallucination detection. I suggest the following changes.

* **Add a clean-subset analysis.** In addition to the main analysis on existing benchmarks, it would be useful to report results after excluding clearly ambiguous prompts, test-case errors, and cases where the benchmark accepts only one of several valid library-use patterns. This is not necessary for the paper’s empirical goal, but it would help separate intrinsic library hallucinations from benchmark-induced failures.

* **Discuss the dependence on tool-specific knowledge sources.** The comparison between Grammar, Mypy, Pyright, and o3-mini partly reflects differences in their underlying knowledge sources, such as docstrings, type stubs, type inference, and LLM prior knowledge. A more explicit discussion of this issue would make the comparison easier to interpret.

* **Provide more representative examples of false positives and false negatives.** The paper would be clearer if it included additional examples showing why a hallucination was caught for the wrong reason, why a static tool missed it, and why some cases are infeasible for static analysis.

---

> ### Author Response · Authors · 2026-06-19
>
> Thank you for the thoughtful and constructive feedback. We have addressed all requested changes in the revised manuscript. Below we detail our responses point by point.
>
> **W1. Clean-subset analysis**
> We agree this is a valuable addition and have included a new subsection **Section 5.4 "Validity Against Benchmark Noise"** in the revised manuscript.
>
> "We ran the analysis on Imaginary Features (IF) across all three experiments for the subset of prompts identified as valid, which translated into removing 3% of DS-1000, 7.7% of ODEX, and 6.1% of BigCodeBench. We then computed the difference between full-set and valid-subset metrics for detection (ΔIF_R), repair (ΔR_IF), and mitigation (ΔIF_%).
>
> We conducted a Spearman correlation for each experiment and benchmark between the original metric and the valid subset metric and found that the ranking of models' performance was significantly preserved at 0.001, indicating it was not affected by benchmark noise. For detection, the average correlation was 0.98 with average ΔIF_R = 0.02, and for constrained decoding, the average correlation was 0.99 with average ΔIF_%  =0.60; Figure 12 in the Appendix shows an almost perfect correlation between these two metrics, while in percentage removing the subset will change the results by less than 1%. In repair, we have a different case with a lower average p=0.88, significant at 0.001. As seen in Figure 3, across the three benchmarks, the valid subset R_IF is lower than the full set of R_I, with an average ΔR_IF =- 4.1, suggesting that approximately 4% of repair failures cannot be fixed due to the benchmark rather than the model."
>
> **W2. Dependence on tool-specific knowledge sources**
> We agree that making this distinction explicit improves interpretability. In the revised manuscript, we replaced Table 2 with Figure 1, which is breakdown of IF recall (IF_R) by error type, and expanded **Section 5.1** with the following analysis:
>
> "We can see that the performance of each tool by error type reflects differences in its underlying knowledge sources. Rule checkers such as mypy and grammar perform best on ImportError and ModuleNotFound since they rely on syntax rules and import registries that only require verifying functions in the library. Pyright is broader by incorporating type stubs and inference from typed codebases, which can help resolve type mutations, improving its effectiveness against TypeError and AttributeError. o3-mini performs poorly on the simpler case of distinguishing imaginary solves for imports; however, it can serve as an inference tool for type mutation. This shows that static analysis is a cheap and reliable deterministic tool for identifying a hallucinated library, where LLMs' prior knowledge is ungrounded."
>
> **W3. More examples on false positives and false negatives**
> We have reorganized **Section 5.1.1** into three dedicated subsections to address this directly: "Annotation Reliability and Validation", "Detection Boundaries of Static Methods", and "Type Annotations Remain a Challenge for Static Analysis". Each subsection now includes concrete examples explicitly framed in terms of false positives and false negatives, covering cases caught for the wrong reason, cases missed by static tools, and cases that are structurally infeasible for static analysis.
>
> We have also renamed two sections for clarity: **Section 5.1.1** is now **"Investigating Potential and Overcoming False Negatives"**, and **Section 5.1.2** is now **"Causes of Hallucinations and Identifying False Positives"**.
>
> **Broader Impact. Security implications of library hallucinations**
> We agree this connection deserves explicit treatment and have extended the discussion section accordingly: "Unlike approaches that sample higher-quality tokens without explicitly addressing detection and mitigation simultaneously, this work provides a framework that directly targets both, though fully eliminating hallucinations remains an open challenge for the field (Tanzil et al., 2024). However, as shown in Figure 1, static analysis tools and grammars are useful for detecting hallucinated packages before execution and can prevent malicious packages from running on users' systems (Spracklen et al., 2025; Krishna et al., 2025)."

---

### Review · Reviewer_6qnm · 2026-06-10

**Summary Of Contributions:**

This paper is an empirical study of static analysis methods for detecting and mitigating library hallucinations in LLM-generated Python code. The hallucinations are defined as incorrect uses of external library/API facts. The paper evaluates grammar, Mypy, Pyright, and an LLM-judge across six code-generation models and three NL-to-code benchmarks. Results show that Pyright can detect a meaningful portion of hallucinations, while grammar-based methods are precise but have low recall. The paper further studies repair with o3-mini and mitigation through grammar-constrained decoding, finding that repair improves execution rate but does not remove hallucination.

**Additional Comments:**

**Strength:**

- The paper targets an important and practical problem in LLM code generation: hallucinated library usage.
- The empirical evaluation covers multiple benchmarks, multiple generation models, and several detection/mitigation methods, making the findings and observations general.

**Weakness:**

- Some implementation and annotation details are insufficiently explained. The grammar-construction pipeline is only described at a high level in the main text.
- The main results report aggregate IF/IFR metrics, but do not provide detailed type breakdowns, e.g., attribute, import, type, and module-not-found errors. A more detailed breakdown would help understand what kinds of library hallucinations each static method can or cannot detect.
- The “upper bound” estimation is from subjective judgment. The decision rules and examples should be made clearer in the main paper.
- Inconsistent or potentially misleading statements of the result. First, Section 5.1 says grammar recall on imaginary features ranges from 8.1% to 40%, but Table 2 reports grammar IFR values ranging roughly from 0.06 to 0.20. Second, the text claims that Mypy and Pyright have “higher recall than precision,” but Table 1 contains several places where precision is higher than recall.

**Audience:**

Yes

**Audience Explanation:**

The paper studies library hallucinations in LLM-generated code, which is a timely and practically important problem. Its findings on the strengths and limitations of static analysis, repair, and grammar-constrained decoding would be interesting to researchers working in LLM-code generation directions.

**Claims And Evidence:**

Yes

**Claims Explanation:**

The main claims are supported by experiments across multiple LLMs, benchmarks, and detection/mitigation methods. However, there are some inconsistencies between the claim and the actual experiment. Please see the last weakness point.

**Requested Changes:**

Please see the weakness section and revise the paper accordingly.

---

> ### Author Response · Authors · 2026-06-19
>
> Thank you for the thoughtful and constructive feedback. We have addressed all requested changes in the revised manuscript. Below, we detail our responses point by point.
>
> **W1. Grammar implementation details**
> We agree that the original description was insufficient. We have expanded the grammar implementation description in the main text **Section 4.1** and added a concrete example in the **Appendix B**, allowing readers to follow the full construction pipeline in detail.
>
> **W2. Error type breakdown by static tool**
> We agree that showing error types rather than aggregate metrics will provide insights into each tool's capabilities. In the revised manuscript, we replaced Table 2 with Figure 1, a breakdown of IF recall by error type, and expanded **Section 5.1** with the following passage:
>
> "We can see that the performance of each tool by error type reflects differences in its underlying knowledge sources. Rule checkers such as mypy and grammar perform best on ImportError and ModuleNotFound since they rely on syntax rules and import registries that only require verifying functions in the library. Pyright is broader by incorporating type stubs and inference from typed codebases, which can help resolve type mutations, improving its effectiveness against TypeError and AttributeError. o3-mini performs poorly on the simpler case of distinguishing imaginary solves for imports; however, it can serve as an inference tool for type mutation. This shows that static analysis is a cheap and reliable deterministic tool for identifying a hallucinated library, where LLMs' prior knowledge is ungrounded."
>
> **W3 Decision rules and examples for the upper bound estimation**
> We agree that the subjectivity of the upper bound estimation required clearer documentation, therefore:
>
> We have extended the reference to Appendix Table 9, which contains annotated examples, and moved it from Section 5.1.1 "Annotation Reliability and Validation" to **Section 5.1.2** "Causes of Hallucinations and Identifying False Positives", as this section is thematically better suited to discuss root cause failures and detection boundaries. It is now described as follows:
>
> "Table 9 in the Appendix describes examples of root cause failures and why these could be detected based on the capability of each detection tool."
>
> **W4. Inconsistent and potentially misleading statements of results**
> Thanks for catching these inconsistencies. We addressed as follows:
> - The discrepancy between the grammar recall range reported in **Section 5.1** (8.1% to 40%) and the IF R values in Table 2 (0.06 to 0.20) was because we reported the raw percentage of cases caught rather than the IF_R (IF recall) shown in the table.
> We have corrected the text to report IF R values consistently throughout.
> - The claim that Mypy and Pyright have "higher recall than precision" was imprecise. The revised text now reads:
>
> "Mypy and Pyright, the off-the-shelf static analysis tools, have very similar results, with higher precision than recall on DS-1000 and BigCode; however, on Odex SOTA models have higher recall than precision, as further discussed in Section 5.1.1."
>
> We also updated **Section 5.1.1** to have this specific explanation:
>
> “In Table 2, we see that on Odex, static tools are capable of detecting 70.5% of hallucinations; this also translates into higher recall observed in Table 1. This is because many cases of Odex do not return anything in the function, resulting in returning None being an easy true positive for the Static tools to identify. However, this is not the case for BigCode, with a more diverse source of hallucinations.”

---

### Review · Reviewer_ysmo · 2026-06-15

**Summary Of Contributions:**

This paper presents an empirical study of using static analysis tools (Mypy, Pyright) and automatically extracted grammars to detect and mitigate library hallucinations in LLM-generated Python code. The authors evaluate six LLMs on three code generation benchmarks (DS‑1000, ODEX, BigCodeBench). They measure detection precision/recall for both general errors and hallucination‑specific errors (imaginary features). They also attempt repair via prompting with tool feedback and mitigate via grammar‑constrained decoding. A manual analysis of 200 hallucination cases establishes an upper bound of static detection (up to 77%) and identifies error categories that static methods cannot address (e.g., ambiguous prompts, test‑case issues, data/logic errors). Key findings: static analysis tools detect many errors but with false positives; detection does not transfer to repair; constrained decoding reduces some hallucinations but can hurt correctness; many apparent hallucinations stem from benchmark underspecification.

Strengths:

Well‑defined, practically relevant problem (library hallucinations in code generation).

Comprehensive empirical setup across multiple LLMs, benchmarks, and detection methods.

Manual annotation and upper‑bound analysis provide valuable insights beyond raw metrics.

Clear identification of blind spots (data/control flow, ambiguous prompts, test‑case limitations).

Weaknesses:

Novelty is limited – static analysis and grammar constraints are well‑known techniques; the main contribution is a systematic evaluation rather than a new method.

The repair experiments show little to no improvement on hallucinations, and the paper does not deeply explore why or propose alternatives.

Constrained decoding results are mixed; the Pass@1 decrease is noted but not thoroughly analysed.

The writing is dense and some tables/figures are hard to interpret without frequent cross‑referencing.

**Audience:**

Yes

**Audience Explanation:**

Yes. The paper addresses a practical problem for developers using LLM code generation: how to detect and mitigate hallucinations involving non‑existent library features. TMLR readers working on code generation, software engineering, or LLM evaluation will find the empirical bounds and failure mode taxonomy valuable. The finding that static analysis can catch up to ~77% of hallucinations (but not all) and that repair does not easily fix them is actionable. The paper also highlights important flaws in current benchmarks (ambiguous prompts, test‑case issues), which is relevant to the benchmarking community.

**Broader Impact Concerns:**

The paper does not raise significant ethical concerns. Its contributions are technical and empirical, aimed at improving the reliability of LLM‑generated code. Potential positive impacts include reducing runtime errors and security risks from hallucinated library calls. There is no discussion of misuse or negative societal impact, which is acceptable given the topic. A brief broader impact statement is not required for this type of empirical study, but the authors could add a sentence acknowledging that improved detection tools may also be used to evaluate and compare LLMs, which could influence model development priorities. No ethical red flags are present.

**Claims And Evidence:**

Yes

**Claims Explanation:**

The evidence is largely convincing:

Detection claims are supported by Table 1 (precision/recall for all errors) and Table 2 (recall on imaginary features). The authors clearly show variability across models and tools.

Upper bound on static analysis (Section 5.1.1, Table 3) is derived from manual annotation with good inter‑annotator agreement (Cohen’s kappa ~0.79), providing a credible estimate.

Repair results (Tables 5, 6) show that while executability improves, hallucination rate does not consistently decrease – a clear and honest finding.

Mitigation via constrained decoding (Table 7) shows reductions in imaginary features on ODEX but mixed results elsewhere, with a plausible explanation (grammar completeness, distribution shift).

Annotation reliability is properly reported (kappa values, Table 9).

The only minor concern is that some comparisons (e.g., o3‑mini baseline) are not directly comparable across all metrics due to different underlying detection logic, but this is acknowledged. Overall, the evidence is accurate and sufficient.

**Requested Changes:**

Clarify the definition of “hallucination” vs. “error” early and consistently (critical).
Section 3 defines hallucination as unfaithfulness to external facts (libraries), but later the metrics sometimes treat all errors as hallucination. Please ensure the distinction is maintained throughout, especially in the detection tables.

Provide more analysis on why constrained decoding sometimes reduces Pass@1 (strengthening).
Table 7 shows Pass@1 drops on DS‑1000 for Gemma2 and Qwen3. A short discussion (beyond docstring incompleteness) would help readers decide when to use this method.

Add a discussion of the practical cost‑benefit trade‑off (strengthening).
The paper reports time overhead (12 seconds extra on average) but does not weigh this against the benefit. A simple cost‑benefit summary (e.g., “to catch X% of hallucinations, it costs Y seconds per request”) would improve practical utility.

Improve table readability (minor).
Tables 1, 2, 5, 6, 7 have dense formatting and many acronyms (OP, OR, IFR, ER, RIF). Please add a legend or footnote explaining these. Also, the baseline row label “o3‑mini” is inconsistently placed in Table 1.

Acknowledge the limitation to Python (minor, already partly done).
The paper focuses on Python; the grammar construction method is Python‑specific. Please explicitly state that generalisation to other languages would require non‑trivial adaptation (as briefly mentioned in Section 4.1, but could be emphasised in conclusion).

Correct small typos (minor).

Table 1: “OP” and “OR” are defined in caption but not expanded in the table; consider using “Precision” and “Recall”.

Section 5.1.1: “we manually inspected a sample of 200 failure cases” – specify whether this is across all benchmarks combined or per benchmark.

---

> ### Author Response · Authors · 2026-06-19
>
> Thank you for the thoughtful and constructive feedback. We have addressed all requested changes in the revised manuscript. Below, we detail our responses point by point.
>
> **W1. Consistency on hallucination definition** We inspect every occurrence of "hallucination/imaginary features" and "error" across the main text, tables, and figures to ensure consistent usage.
>
> In **Section 3 "Defining Code Hallucinations"**, we strengthened the definition to make explicit that hallucinations refer specifically to unfaithfulness to external library facts, as opposed to general code errors such as syntax or logic mistakes.
>
> In **Section 5.1 “Detection”**, reporting hallucinations and all code errors provides a complete picture of tool behavior, as tools will inevitably flag some non-hallucination errors. Precision and recall over all errors in Table 1 capture the overall detection profile (which is now explicitly stated in the caption table), while Figure 1 isolates detection performance on hallucinations only (IF_R). To make this explicit, we extended **Section 5 "Metrics"**:
>
> "For detection, we consider precision and recall on (1) all errors, and (2) hallucination-specific errors, which we call Imaginary Features (IF). We quantify Imaginary Features Recall (IFR) as...”
> Each equation now explicitly states which subset is being reported, thereby avoiding ambiguity.
>
> **W2. Discussion on why constrained decoding sometimes reduces Pass@1** We have extended the discussion to point to Figure 2, which supports a second explanation beyond docstring incompleteness in **Section 5.3, paragraph 2**:
>
> "Another case is when valid alternative solutions fall outside the LLM's sampling distribution; even correct candidates in the grammar might not appear in the LLM's pretraining data. Figure 2 shows an example of how the grammar can help. Without constrained decoding, the LLM produced an imaginary parameter use_line_collection; with constrained decoding, it generates valid parameters but not the correct answer, suggesting the issue is distributional rather than grammatical."
>
> **W3. Cost-benefit summary on constrained decoding** We agree that reporting time overhead alone is insufficient for practitioners to assess the utility of constrained decoding. We have updated the section with a concrete cost-benefit summary in **Section 5.3, paragraph 3**:
>
> "Odex averaged 1.56 resamplings per request (one to three additional samplings) versus 3.32 for DS-1000 (three to eight additional samplings). On Odex, the 12-second overhead reduces hallucinations by 0.7 points in IF_%; doubling the overhead in DS-1000 yields a larger reduction of 3.7 points."
>
> **W4. Table readability**
>
> We have revisited all tables and updated their captions, particularly **Table 6**, to improve clarity. We have also added legends and footnotes explaining the acronyms used throughout, including IF_R, R_IF, IF_%, and others, and replaced OP and OR in **Table 1** with P (Precision) and R (Recall) for consistency with the caption definitions.
>
> **W5. Generalisation to other languages**  We have added the following statement to the **conclusion** to explicitly acknowledge the Python-specific scope of the grammar construction method:
>
> "This work provides the first empirical analysis of static analysis and grammar-constrained decoding methods for detecting and mitigating library hallucinations in NL-to-code settings. Our evaluation is limited to Python, a choice made because it is the dominant language for execution-based benchmarks. While the framework is general, the grammar construction method for other languages would require non-trivial adaptation, as it relies on the languages' formal grammars and on library inspection...”
>
> **W6 . Clarification on the 200 failure cases sample** We have updated **Section 5.1.1 "Annotation reliability and validation"**  to clarify the sampling procedure:
>
> "To identify the upper bound on performance of static methods, we manually inspected a sample of 200 failure cases in the subset of Imaginary Features per benchmark (600 in total), with an even sample size across four models (claude-3, gpt-3.5, gpt-4, and Granite 3B)."
>
> **Broader Impact. Detection tools as a form of LLM evaluation** We agree this connection is worth making explicit. We have extended the final paragraph of the **discussion** as follows:
>
> "Beyond mitigation, such detection tools may also serve as evaluation instruments for comparing LLMs' tendencies to hallucinate library features, which can potentially influence the development of code models. Together, our findings suggest that future directions for this problem may benefit from methods beyond the output-level interventions tested in this study, static analysis, and grammar-constrained decoding, and more towards understanding how these hallucinations originate in the first place."